# PoI: Pixel of Interest for Novel View Synthesis Assisted Scene Coordinate Regression

## Abstract

The task of estimating camera poses can be enhanced through novel view synthesis techniques such as NeRF and Gaussian Splatting to increase the diversity and extension of training data. However, these techniques often produce rendered images with issues like blurring and ghosting, which compromise their reliability. These issues become particularly pronounced for Scene Coordinate Regression (SCR) methods, which estimate 3D coordinates at the pixel level. To mitigate the problems associated with unreliable rendered images, we introduce a novel filtering approach, which selectively extracts well-rendered pixels while discarding the inferior ones. The threshold of this filter is adaptively determined by the real-time reprojection loss recorded by the SCR models during training. Building on this filtering technique, we also develop a new strategy to improve scene coordinate regression using sparse inputs, drawing on successful applications of sparse input techniques in novel view synthesis. Our experimental results validate the effectiveness of our method, demonstrating the state-of-the-art performance on both indoor and outdoor datasets.

## 1 Introduction

Visual localization, also known as camera relocalization, is a fundamental task in computer vision that involves estimating the 6-degree-of-freedom (6DOF) camera poses within a known scene based on input images. This task plays a crucial role in Simultaneous Localization and Mapping (SLAM) (Izadi et al., 2011; Mur-Artal et al., 2015; Dai et al., 2017; Tang & Tan, 2018) and has significant applications in areas such as autonomous driving, robotics, and virtual reality.

Traditional methods for camera relocalization can be categorized into two main types: Camera Pose Regression (CPR) methods (Chen et al., 2021; Ng et al., 2021; Purkait et al., 2018; Taira et al., 2018; Moreau et al., 2022a;b; Chen et al., 2022) and Scene Coordinate Regression (SCR) methods (Brachmann & Rother, 2021; Brachmann et al., 2017; Brachmann & Rother, 2019; Shotton et al., 2013; Valentin et al., 2015; Brachmann et al., 2023). Between these, SCR frameworks are particularly favored due to their higher accuracy. However, both approaches require stringent sampling density of training data to ensure reliable pose estimations for arbitrary images captured within a specific scene. Manually collecting a sufficient number of training images is a time-consuming process, and obtaining the corresponding camera pose labels presents further difficulties.

In light of this, the CPR-based methods try to enrich the training set with synthetic data rendered by novel view synthesis (NVS) techniques. For example, LENS (Moreau et al., 2022b) employs NeRF to render average sampled novel views, thereby augmenting the training dataset and treating these synthetic images similarly to real data without additional processing. Similarly, DFNet (Chen et al., 2022) utilizes NeRF-W (Martin-Brualla et al., 2021) for NVS and features a cross-domain design that helps to minimize the discrepancies between synthetic and query images, effectively bridging the gap between the two domains.

Currently, there is no similar research within the SCR framework, and we raise the question of whether SCR-based pipelines can also benefit from synthetic images. To this end, we attempt to apply NVS for data augmentation within the SCR framework. Nevertheless, we found that SCR methods, which rely on precise pixel-to-pixel (N2N) predictions, are particularly vulnerable to the quality of rendered images. This contrasts with CPR methods, which involve pixel-to-pose predictions and are less affected by image quality. As shown in the right section of Figure 1, after

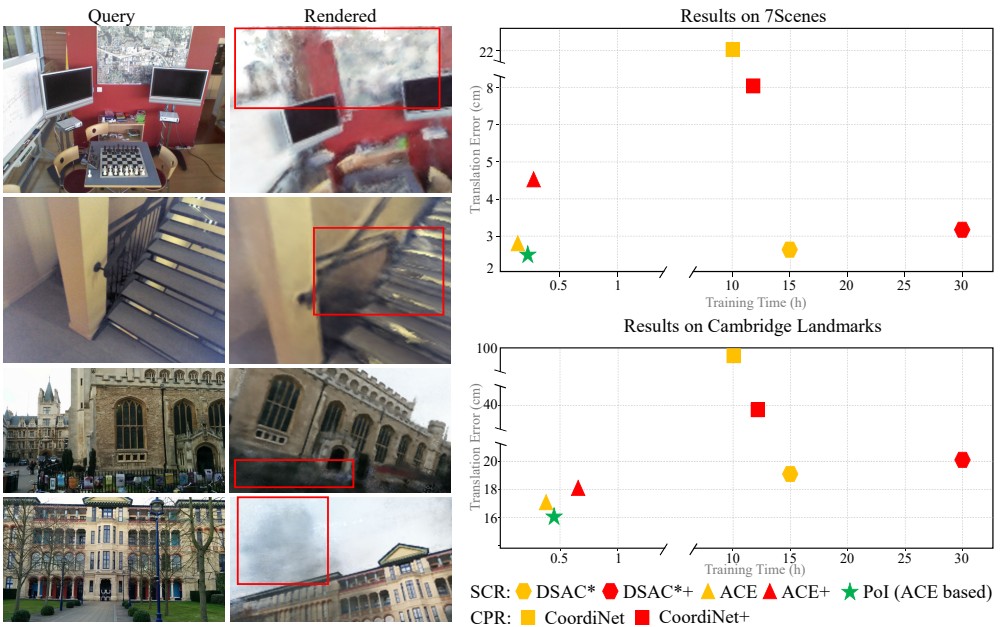

Figure 1: **Left**: Comparison of query and rendered images of the dataset 7Scenes and Cambridge Landmarks, revealing uneven rendering quality within frames, with some parts clear and others blurry or ghosted; **Right**: Translation error versus training time, where "CoodiNet+" means using rendered images as query images for CPR method CoodiNet (we use LENS in this case); "DSAC*+" and "ACE+" denote the method combines NVS-rendered images and query images as training data for SCR method DSAC* and ACE. "PoI" denotes our method; We can see that directly adding rendered data to the training set will increase training time to some extent, but performance will decrease for the SCR method. On the other hand, our PoI approach can improve the performance with an acceptable time increase.

expanding the training dataset with synthetic data from NVS, the CPR method shows significant improvement, while SCR performance declines, accompanied by a notable increase in training time. Directly training the SCR model with raw rendered images proves less effective than CPR methods and may even result in model collapse if the proportion of rendered images is excessively high.

To tackle this issue, we design a portable pixel of interest (PoI) module that serves as an effective filter for synthetic clues. Specifically, the 3D-to-2D projection error of each pixel is employed as a criterion for whether the point is retained or not, and use a rough to precise threshold setting for screening at different training stages. As the training progresses, PoI gradually removes poorly rendered pixels and further leverages the remaining points alongside real data to train the network.

Moreover, we propose a coarse-to-fine variant of PoI to address the challenges of visual localization in extreme scenarios, especially where training data is limited. In the coarse stage, PoI receives all available synthetic data as inputs, progressively training the coarse model with valid rendered pixels. Following this, we fine-tune the coarse model using sparse real pixels. This method enables our PoI variant to efficiently leverage sparse inputs while ensuring strong pose estimation performance, even in difficult conditions.

The main contributions of our work are summarized as follows:

- We introduce PoI, a pixel-level filter designed to eliminate poorly rendered pixels for effective training data augmentation.

- We present an innovative approach to tackle scene coordinate regression from sparse inputs.

- Our method achieves state-of-the-art performance on both indoor and outdoor datasets.

## 2 RELATED WORK

**Camera Pose Regression** The CPR methods, i.e., to regress the camera pose from the given image directly, are the most naive ideas and most widely used in learning-based methods (Kendall et al., 2015; Brachmann et al., 2016; Brahmbhatt et al., 2018; Melekhov et al., 2017; Radwan et al., 2018; Wang et al., 2020; Hu et al., 2020; Arnold et al., 2022; Chen et al., 2022; Shavit & Keller, 2022). The most straightforward method implicitly uses CNN layers or MLP to represent the image-to-pose correspondence. PoseNet (Kendall et al., 2015) first proposes this using pre-trained GoogLeNet as the feature extractor. Then, several works focus on improving CPR through additional modules. Geomapnet (Brahmbhatt et al., 2018) estimates the absolute camera poses and the relative poses between adjacent frames. AtLoc (Wang et al., 2020) uses a self-attention module to extract salient features from the image. Vlocnet++ (Radwan et al., 2018) adds a semantic module to solve the dynamic scene and improve the robustness for blockings and blurs. Marepo (Chen et al., 2024a) first regresses the scene-specific geometry from the input images and then estimates the camera pose using a scene-agnostic transformer. The CPR method has achieved excellent efficiency and simplification of the framework, but there is still room for improvement in accuracy.

**Scene Coordinate Regression** Recently, the SCR methods (Shotton et al., 2013; Brachmann et al., 2017; Brachmann & Rother, 2018; 2019; Massiceti et al., 2017; Li et al., 2018; Brachmann & Rother, 2021) achieve better performance in terms of the accuracy compared with the CPR methods. The SCR method aims to estimate the coordinates of the points in 3D scenes instead of relying on the feature extractor to find salient descriptors, as in CPR methods. SCR was initially proposed using the random forest for RGB-D images (Shotton et al., 2013). Recently, estimating scene coordinates through RGB input has been widely studied. ForestNet(Massiceti et al., 2017) compares the benefits of Random Forest (RF) and Neural Networks in evaluating the scene coordinate and camera poses. ForestNet also proposes a novel method to initiate the neural network from an RF. DSAC (Brachmann et al., 2017), DSAC++ (Brachmann & Rother, 2018) devise a differentiable RANSAC, and thus the SCR method can be trained end-to-end. ESAC (Brachmann & Rother, 2019) uses a mixture of expert models (i.e., a gating network) to decide which domain the query belongs to, and then the complex SCR task can be split into simpler ones. DSAC* (Brachmann & Rother, 2021) extends the previous works to applications using RGB-D or RGB images, with/without the 3D models. This means that in the minimal case, only RGB images will be used as the input to DSAC*, just like most CPR methods. More information about the 3D structure will be utilized for most SCR methods than CPR ones. However, approaches like DSAC* can achieve more accurate estimations even if the input is the same as the CPR method. ACE (Brachmann et al., 2023) and GLACE (Wang et al., 2024) abandon the time-consuming end-to-end supervision module and shuffle all pixels of the scene to improve training efficiency. ACE and GLACE use only RGBs without extra 3D geometry information and achieve comparable accuracy compared with former methods.

Despite the progress of CPR and SCR methods, both methods still have great problems in data collection and labeling. Therefore, efficient data collection and labeling methods or alternatives with similar effects are needed.

**Novel view synthesis (NVS) for pose estimation** A major challenge for visual localization methods is collecting appropriate photos to cover the entire scene. Essentially, the number and distribution of image sets for training are difficult to decide. For example, most outdoor scenes collect data along roads, such as Cambridge landmarks (Kendall et al., 2015). For indoor datasets (such as the 7Scenes (Shotton et al., 2013) dataset), all translations and orientations within the scene are considered.

To fulfill the diverse requirements of data collecting, some works try to use more flexible NVS to render synthetic views instead of collecting extra data (Chen et al., 2021; Ng et al., 2021; Purkait et al., 2018; Taira et al., 2018; Moreau et al., 2022b; Chen et al., 2022), where NVS is the method to render synthetic images from the camera poses, which can verify the accuracy of 3D reconstruction, especially for implicit reconstruction methods like NeRF (Mildenhall et al., 2021), and Gaussian Splatting (Kerbl et al., 2023). INeRF (Yen-Chen et al., 2021) applies an inverted NeRF to optimize the estimated pose through color residual between rendered and observed images. However, the initially estimated poses are significant in guaranteeing the convergence of outputs. LENS (Moreau et al., 2022b) samples the poses uniformly all over the area and trains a NeRF-W (Martin-Brualla et al., 2021) to render the synthetic images. Then, rendered images and poses work as the additional

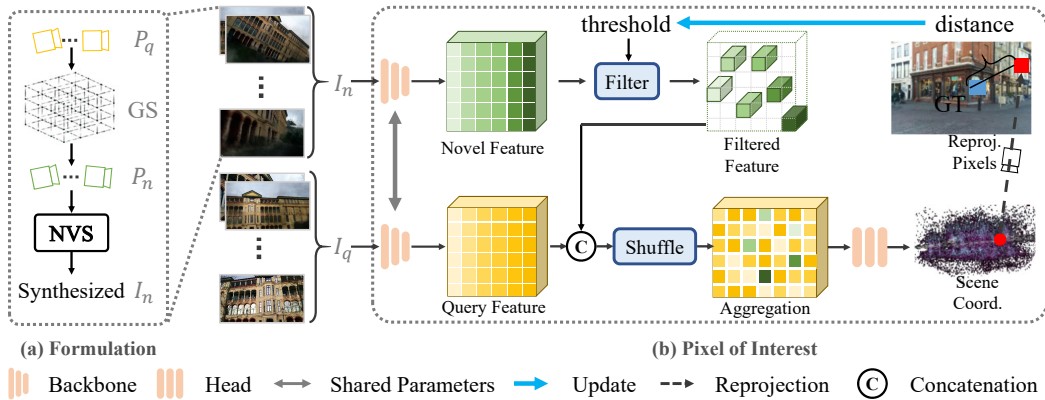

Figure 2: Pipeline of our proposed methods: **(a) data formulation**: We first sample a group of synthesized camera poses $P_n$ according to the query training pose $P_q$ using 'GS' (grid sampling). Then, we render the synthesized views $I_n$ based on the sampled poses $P_n$ through the novel view synthesis model. **(b) architecture of PoI**: First, a pre-trained scene-irrelevant backbone is applied to extract the features of the input query photos $I_q$ and the synthesized novel images $I_n$. Then, the filter is applied to the features of the rendered images and gets the features of interest. After that, we combine the query features with the filtered novel features and shuffle the pixel-aligned features to get the aggregation. Finally, we estimate the scene coordinates of the pixels using a scene-specific Head. The filtering algorithm is designed based on the re-projection error of the estimated scene coordinates.

training data for the pose regression network. The limitation of LENS lies in the costly offline computation for dense samples. DFNet uses direct feature matching between observed and synthetic images generated by histogram-assisted NeRF. The feature match approach is proposed to extract observed or generated images' cross-domain information. All these methods combine the NVS module and the CPR module to optimize the performance of the absolute pose estimation of the photos.

Unlike the former methods, we propose using an SCR rather than the CPR method with proposed NVS rules to improve the camera pose estimation. First, we design novel pose sampling methods to meet multiple requirements of different datasets. To address the problem of varying lighting conditions, we adopt the NeRF-W as the baseline to sample new views of multiple lightings for each sampled pose in outdoor datasets. Second, we propose a pixel filter to remove bad pixels in rendered images and use captured frames and remaining rendered pixels to improve the estimation.

## 3 PRELIMINARY

### 3.1 NVS MODELS USED IN OUR APPROACH

NeRF uses camera poses and the intrinsic matrix to project rays from the pixels of the 2D images into 3D spaces. Then, it will sample a certain number of points from each ray. The color and volume density for each 3D point would be estimated with the supervision of rendering loss: the mean square error between the query and the rendered pixel colors. The overall process can be expressed in this equation:

$$\hat{C}_r = \mathcal{R}(r, c, \sigma) = \sum_{k=1}^{K} T(t_k)\alpha(\sigma(t_k)\delta_k)c(t_k) \tag{1}$$

where $T(t_k) = exp(-\sum_{t_{k'}=1}^{k-1} \sigma(t_{k'})\delta_{k'})$.

This paper uses NeRF-W (Martin-Brualla et al., 2021) as the baseline for novel view synthesis. NeRF-W is designed to render novel views through the unstructured collection of outdoor images.

The main challenge of this situation is the change in illumination conditions and the occlusion of dynamic objects. The same situation also exists in the camera relocalization problem. For example, if we ignore illumination conditions, we cannot estimate the photos taken in the morning using the model trained through the data collected at night. To solve this problem, NeRF-W uses additional appearance embedding and dynamic embedding as input for the MLPs. This enables us to choose the appearance condition while rendering novel views. Moreover, NeRF-W can wipe out the dynamic objects from the scene with the predicted uncertainty. The improvement of NeRF-W can be expressed as:

$$
\begin{aligned}
\hat{C}_r =& \mathcal{R}(r, c_i, \sigma) \\
c_i(t) =& MLP_\theta(z(t), \gamma_d(d), \ell_i^{(a)}) \\
\hat{C}_r =& \sum_{k=1}^{K} T(t_k)(\alpha(\sigma(t_k)\delta_k)c_i(t_k) + \alpha(\sigma_i^\tau(t_k)\delta_k)c_i^\tau(t_k))
\end{aligned}
\tag{2}
$$

where $T_i(t_k) = \exp(-\sum_{t_{k'}=1}^{k-1}(\sigma(t_{k'}) + \sigma_i^{(\tau)}(t_{k'}))\delta_{k'})$. $i$ denotes the image index; the static density is irrelevant to $i$, but the color is related to $i$ because of the appearance change. $\sigma_i^{(\tau)}$ represents the density of the dynamic model, which is also related to $i$, and NeRF-W uses a dynamic embedding based on $i$ as input. By using this method, we can reliably render novel views of controllable illumination conditions and mask the dynamic objects from the results.

## 4 METHOD

Overall, the pipeline of our proposed method is shown in Figure 2. For the input query images $I_q$, and corresponding camera poses $P_q$, we first sample the novel camera pose $P_n$ using Grid Sampling (GS). Then we render novel views $I_n$ using NeRF-W. Finally, we use PoI to estimate the scene coordinates through the input $I_q$, $I_n$. During test time, we use PNP-based Ransac to infer the camera poses from the scene coordinates.

The following part of this chapter is arranged as follows:

- Chapter 4.1 elaborates on the details of the proposed method: PoI;
- Chapter 4.2 introduces using PoI as a plugin to non-end-to-end SCR.
- Chapter 4.3 explains the variant of PoI in extreme cases of sparse input.

### 4.1 PIXEL OF INTEREST (POI)

To use rendered images as an auxiliary input for camera pose estimation, most existing methods estimate the scene coordinates of all pixels (or downsampled pixels) of the rendered image without considering the difference in rendering quality of these pixels, which greatly increases the time and resource cost of training and reduces the effectiveness of auxiliary data. To improve training efficiency and effectiveness, we are thinking of reducing the number of rendered pixels compared with query images for training. Considering that the Nerf-based reconstruction method predicts the target RGB pixel-wise without cross-pixel guidance, the rendering quality of different pixels from the same image would be independent. So, if we reduce the rendered images frame-wise, some well-rendered pixels of the discarded images would also be removed. To address this problem, we propose a method that finds the well-rendered pixels of the frame: pixels of interest.

The architecture of PoI is illustrated in Figure 2.(b). In order to filter out poorly rendered pixels, we need a method to obtain pixel-level feature supervision instead of frame-level feature map supervision. We use the pre-trained scene-agnostic convolutional network from Ace (Brachmann et al., 2023) as our backbone to obtain frame-level feature maps, and we would fix the parameters of this backbone network during our entire training process. We input the query images $I_q$ and the synthesized images $I_n$ into the backbone and get query features and novel features. We keep all of the query features, while we use a filtering algorithm to extract features of interest (FOI) from the novel features. The filtering algorithm can be divided into two parts: First, we randomly sample the novel

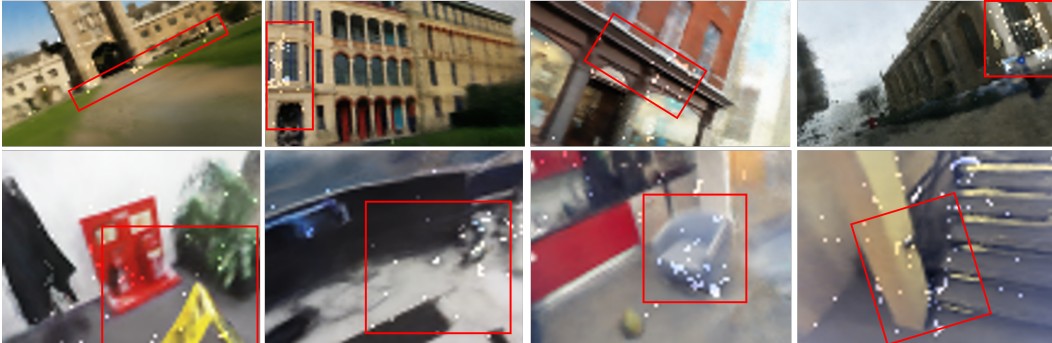

Figure 3: An example of the results of PoI in dataset 7Scenes and Cambridge Landmarks. To highlight the determined pixels of interest, we scale up the 'Value' (V) of the HSV representation of the images.

features at a certain ratio. We want to use more features from query images and fewer features from rendered images to avoid the collapse of the model caused by low-quality rendered pixels, so we set the ratio to 0.1 in our experiments; This ratio is related to the performance of NVS, we may choose a bigger ratio with a better NVS. Second, the filtering threshold is set according to the reprojection loss of these pixels (the distance between GT planar coordinates and estimated reprojected planar coordinates). We periodically rule out the outlier pixels during training. The novel features of outlier prediction will be removed by the filter. The remaining features are the so-called FoI, and the corresponding pixels of FoI are the so-called PoI. Figure 3 shows an example of the results of PoI on 7Scenes and Cambridge Landmarks. After filtering, we combine and shuffle the features $F$ and FOI and put them into the scene-specific MLP Head to estimate the scene coordinates.

It is worth mentioning that we have set a dynamic weight for the loss of rendering pixels. Because at the early step of training, we want the model to converge quickly. After determining the PoI, we gradually reduce the weight of the loss of PoI from 1 to 0.01, while for the pixels from query images, we set the weight to 1 during the whole training process.

$$\mathcal{L} = \begin{cases} \mathcal{L}_{rep}^{query}(i), & \text{if } i \in T \\ \tilde{\omega} \times \mathcal{L}_{rep}^{poi}(i), & \text{if } i \in PoI \end{cases}$$

$$\tilde{\omega} = \omega_{max} - \frac{I_{iter}}{N_{iter}}(\omega_{max} - \omega_{min})$$

(3)

where T denotes traing data, $\tilde{\omega}$ denotes the dynamic weight of PoI loss changing from $\omega_{max}$ (set 1) to $\omega_{min}$ (set 0.01). $I_{iter}$ denotes the current iteration number and $N_{iter}$ is the total iterations. All rendering data is initially set as PoI. As the training progresses, we rule out outlier prediction points from PoI. At the end of the training, the choice of PoI and the loss weight of PoI are fixed.

In PoI, we would sample novel camera poses and render the corresponding images according to the images and the corresponding camera poses from the training set. In existing novel view synthesis supported visual localization, we usually have to balance the novel poses' diversity and the images' overall rendering quality. However, we do not need an overall well-rendered image in the PoI task because of the pixel-level optimization and filtering algorithm. Therefore, we should try to expand the diversity of novel poses. We use a unified sampling method for camera pose translation: grid sampling. The boundaries of the grid are calculated based on the camera pose of the training data; that is, the maximum and minimum values of the grid (x, y, z) are determined by the maximum and minimum values of the translations of all camera poses. We add a random perturbation to the rotation. The original rotation of each grid starts from the closest camera pose to that grid in the training data.

### 4.2 POI AS A PLUG-AND-PLAY MODULE IN NON-END-TO-END SCR METHODS

For the end-to-end SCR approach, PoI is difficult to use as a plug-and-play module. First of all, end-to-end SCR methods use image-level loss as supervision and have no pixel-level performance, which makes them unusable for PoI. Furthermore, even for two-stage SCR methods (init+e2e) like DSAC*, all pixels of the same image should be supervised within one iteration in the e2e stage. If we filter out some pixels of the rendered images, aligning the rendered features and designing a differentiable RANSAC algorithm is difficult. Finally, pixel-wise shuffling (which is difficult to achieve in e2e) is also an important factor, without shuffling, poorly rendered pixels are more likely to appear in a batch of data. As a result, the network is more likely to get stuck in a local minimum.

For non-end-to-end training methods like ACE and GLACE, the difference is that we can easily shuffle pixels from all rendered images and query images because the supervision relies only on the camera intrinsics and the planar coordinates of each pixel without further requirements of per-frame joint supervision.

Take the GLACE as an example; our PoI could also be used as in Figure 2.(b); the difference is that the backbone should be replaced. We use the global encoder from GLACE to extract the same dimension of global features as the ACE features. We add the global feature and the ACE feature together and get the target feature maps. The following procedure remains unchanged.

### 4.3 SPARSE INPUT

Sparse input visual localization is a challenging task since both CPR and SCR are not good at estimating unseen parts of the scene because the regression models are trained only from RGB with weak geometric constraints. However, with the help of sparse-view-NVS, we would obtain enough novel views. The challenge is that rendered frames from sparse-view-NVS are generally of lower quality compared with those from dense-view-NVS. Since our PoI method can make good use of rendered images, it could be used to solve sparse input visual localization problems.

In this case, the size of rendered data will be far larger than real data. If we still use the PoI method, the implicit neural map will be mainly contributed by rendered pixels. The accuracy will be influenced in this case. To address this problem, we propose a coarse-to-fine training approach. In the coarse stage, we use the same setting as in PoI; the only difference is that all training data is rendered images. So, the filter is applied to all rendered pixels which we call the Self-pruning step. In this step, in order to leave adequate pixels for training, we raise the filter's threshold (for reprojection errors). We get a coarse model after self-pruning training. In the refinement stage, we fine-tuned the mapping model using real data and the remaining rendered data; we set the learning rate to lower than that of the coarse stage throughout the fine-tuning process. In this step, all pixels are put into the model without filtering.

We finally get the finetuned model and experiment on both indoor and outdoor datasets. The results and implementation details can be found in chapter 5.3.

## 5 EXPERIMENT

### 5.1 IMPLEMENTATION DETAILS

**Dataset** We evaluate the performance of our approaches on two public datasets, Microsoft 7Scenes (Shotton et al., 2013) and Cambridge Landmarks (Kendall et al., 2015). 7Scenes dataset is a collection of RGB-D camera frames consisting of 7 different indoor scenes. Camera tracks are obtained with a KinectFusion system. Cambridge Landmarks include five large-scale outdoor scenes taken around Cambridge University using structure from motion technique to extract the ground truth labels of camera poses.

Our network takes RGB images and the pose labels as input without using the depth information from the 7Scenes dataset or the reconstruction information from the Cambridge Landmarks dataset. We take the original resolution for the RGB images to make an accurate pose estimation.

All data from the training directory of both datasets is used for training the basic PoI training. To save time and computing resources, we do pose sampling and synthesis of new views offline and

Table 1: Median errors of camera pose regression methods and scene coordinate regression methods on the 7Scenes dataset (Shotton et al., 2013). We **bold** the best result for group 'SCR' and group 'SCR w/ glob' seperately.

| | Method | Scenes | | | | | | | Avg. (cm/) |
| --- | --- | --- | --- | --- | --- | --- | --- | --- | --- |
| | | Chess | Fire | Heads | Office | Pumpkin | Kitchen | Stairs | |
| CPR | PoseNet15 | 32/8.12 | 47/14.40 | 29/12.00 | 48/7.68 | 47/8.42 | 59/8.64 | 47/13.80 | 44/10.40 |
| | PoseNet17(geo) | 13/4.48 | 27/11.30 | 17/13.00 | 19/5.55 | 26/4.75 | 23/5.35 | 35/12.40 | 23/8.12 |
| | MapNet | 8/3.25 | 27/11.69 | 18/13.25 | 17/5.15 | 22/4.02 | 23/4.93 | 30/12.08 | 21/7.77 |
| | Hourglass | 15/6.17 | 27/10.84 | 19/11.63 | 21/8.48 | 25/7.01 | 27/10.15 | 29/12.46 | 23/9.53 |
| | LSTM-Pose | 24/5.77 | 34/11.90 | 21/13.70 | 30/8.08 | 33/7.00 | 37/8.83 | 40/13.70 | 31/9.85 |
| | Atloc | 10/4.07 | 25/11.40 | 16/11.80 | 17/5.34 | 21/4.37 | 23/5.42 | 26/10.50 | 20/7.56 |
| | Direct-PN | 10/3.52 | 27/8.66 | 17/13.10 | 16/5.96 | 19/3.85 | 22/5.13 | 32/10.60 | 20/7.26 |
| | GRNet | 8/2.82 | 26/8.94 | 17/11.41 | 18/5.08 | 15/2.77 | 25/4.48 | 23/8.78 | 19/6.33 |
| | ORGMapNet | 9/3.60 | 26/9.49 | 15/12.81 | 20/4.96 | 18/5.04 | 22/5.68 | 27/9.54 | 20/7.30 |
| | LENS | 3/1.30 | 10/3.70 | 7/5.80 | 7/1.90 | 8/2.20 | 9/2.20 | 14/3.60 | 8/3.00 |
| | DFNet | 4/1.48 | 4/2.16 | 3/1.82 | 7/2.01 | 9/2.26 | 9/2.42 | 14/3.31 | 7/2.21 |
| | Marepo | 2.1/1.24 | 2.3/1.39 | 1.8/2.03 | 2.8/1.26 | 3.5/1.48 | 4.2/1.71 | 5.6/1.67 | 3.2/1.54 |
| SCR | DSAC* | 1.9/1.1 | 1.9/1.2 | 1.1/1.8 | 2.6/1.2 | **4.2/1.4** | **3.0/1.7** | 4.1/1.4 | 2.7/1.4 |
| | ACE | **1.9/0.7** | 2.0/0.9 | 1.0/0.7 | 2.7/0.8 | 4.4/1.1 | 4.2/1.3 | 3.8/1.2 | 2.9/0.8 |
| | PoI(ours) | **1.9/0.7** | **1.9/0.9** | **1.0/0.6** | **2.6/0.8** | 4.3/1.1 | 3.9/1.3 | **3.5/1.0** | **2.7/0.8** |
| SCR w/ glob | GLACE | **1.7/0.6** | 1.7/0.8 | **1.1/0.6** | 2.3/0.7 | **3.6/1.0** | **3.4/1.1** | 4.9/1.4 | 2.7/0.8 |
| | GLPoI(ours) | **1.7/0.6** | **1.6/0.7** | 1.1/0.7 | **2.2/0.7** | 3.7/1.0 | **3.4/1.1** | **4.2/1.3** | **2.6/0.8** |

Table 2: Results on Cambridge Landmarks, because of the obvious gap between SCR-based methods and CPR-based methods, we only list SCR-based methods. column 'Mapping time' shows the training time of these methods, and column 'Mapping size' is the memory consumption for saving the parameters of the network. We **bold** the best result for group 'SCR' and group 'SCR w/ glob' separately.

| | Method | Mapping with Depth/Mesh | Mapping Time | Map Size | Scenes | | | | | Avg. (cm/) |
| --- | --- | --- | --- | --- | --- | --- | --- | --- | --- | --- |
| | | | | | King's | Hospital | Shop | Church | Court | |
| FM | AS(SIFT) | No | 35min | 200M | 13/0.2 | 20/0.4 | 4/0.2 | 8/0.3 | 24/0.1 | 14/0.8 |
| | pixLoc | No | 35min | 600M | 14/0.2 | 16/0.3 | 5/0.2 | 10/0.3 | 30/0.1 | 15/0.2 |
| SCR | SANet | Yes | 1min | 260M | 32/0.5 | 32/0.5 | 10/0.5 | 16/0.6 | 328/2 | 84/0.8 |
| | SRC | Yes | 2min | 40M | 39/0.7 | 38/0.5 | 19/1 | 31/1.0 | 81/0.5 | 42/0.7 |
| | DSAC* | No | 15h | 28M | **18/0.3** | **21/0.4** | 5/0.3 | 15/0.6 | 34/0.2 | 19/0.4 |
| | Poker | No | 20min | 16M | **18/0.3** | 25/0.5 | 5/0.3 | **9/0.3** | 28/0.1 | 17/0.3 |
| | PoI (ours) | No | 25min | 16M | **18/0.3** | 23/0.5 | **5/0.2** | **9/0.3** | 27/0.1 | **16/0.3** |
| SCR w/ glob | GLACE | No | 3h | 13M | **19/0.3** | 17/0.4 | **4/0.2** | 9/0.3 | 19/0.1 | 14/0.3 |
| | GLPoI (ours) | No | 3h | 13M | **19/0.3** | **16/0.4** | **4/0.2** | **8/0.3** | **18/0.1** | **13/0.3** |

save the sampled camera poses and the rendered images on disk. During training time, we read this data along with the training set from the disk. We split the training data into two clusters using the camera poses for the scene' kitchen' only. We follow the rule of poker ( a variant of Ace) and train two models with the clusters. During the evaluation, we pick the estimated pose from the model with a more significant number of inlier pixels of the Ransac algorithm. We use one NVIDIA V100 GPU for POI training and use AdamW Loshchilov, 2017 with the learning rate between $5 \times 10^{-4}$ and $5 \times 10^{-3}$. For GLPOI, we use 4 V100 with distributed data-parallel training.

## 5.2 QUANTITATIVE RESULTS

The comparison of median translation and rotation errors between our proposed methods with different absolute camera pose regression methods (at the top), and the scene coordinate regression

Table 3: Median errors of our proposed method with sparse input on 7Scenes and Cambridge dataset.

| Method | 7Scenes | | | Cambridge Landmarks | | |
|---|---|---|---|---|---|---|
| | trans$\downarrow$ | rot$\downarrow$ | $U_{5cm,5\,\mathrm{deg}}\uparrow$ | trans$\downarrow$ | rot$\downarrow$ | $U_{10cm,5\,\mathrm{deg}}\uparrow$ |
| base | 3.7cm | 1.0 | 18.9% | 435cm | 2.2 | 15.7% |
| coarse | 23.1cm | 5.4 | 7.9% | 184cm | 2.2 | 15.8% |
| c2f | **3.5cm** | **0.9** | **36.5%** | **26.9cm** | **0.3** | **20.4%** |

methods (at the bottom) in dataset 7Scenes is shown in Table 1 Generally speaking, scene coordinate regression methods outperform absolute camera pose regression methods in both translations and orientations. DFNet and LENS beat most other approaches within absolute pose regression methods because they use view synthesis methods for data augmentation. Our proposed method outperforms DSAC* and Ace by exploiting the extra information from the rendered novel views. Our 'GLPoI' beats 'GLACE' and achieves the state of the art.

The experiment results on the Cambridge Landmarks datasets are shown in Table 2. Since apparent gaps exist between scene coordinate regression methods and absolute camera pose regression methods, we only list the results of SCR methods and SCR methods with global features. SCR methods include SANet, SRC, DSAC*, Poker (ensembled version of Ace), and our proposed methods. SCR methods with global features include GLACE and the global-feature-version PoI: GLPoI. We come to a similar conclusion as that of 7Scenes. Since our PoI method does not use the time-consuming end-to-end training method like DSAC*, even though we use additional rendered data, it can achieve training efficiency comparable to Ace's.

### 5.3 COARSE-TO-FINE EXPERIMENTS OF SPARSE INPUT

To further evaluate the effectiveness of our method, we do an extra experiment of sparse input as mentioned in Chapter 4.3.

**implemente Details:** We use MVSplat(Chen et al., 2024b) as the sparse NVS model. For datasets like 7Scenes, it takes thousands of images to train a small indoor scene with a scale of only several meters. To simulate the sparse input, We uniformly resample from the input data every 50 frames. For Scene 'heads', we keep only 20 frames for training. For outdoor datasets like Cambridge Landmarks, we split the data into multiple clusters according to the ground truth translations of camera pose (4 in the experiment) and use only one cluster for training.

The numerical results are shown in Table 3, **case 'base'** denotes the sparse input circumstances with the baseline model. **Case 'coarse'** is the method of the self-pruning step of our method only using rendered data; we still use grid sampling as the novel pose sampling method. **Case 'c2f'** denotes the fine-tuned results of our proposed method.

According to the results, we may find that our fine-tuned model can achieve acceptable results with sparse input compared with those using all training data.

## 6 CONCLUSION

In this paper, we propose a pixel-of-interest filter for scene coordinate regression. The filter is designed for non-end-to-end methods which enjoy good converging speed. With the filter, we also design a coarse-to-fine pipeline for sparse input scenarios. We conduct experiments on both indoor and outdoor datasets and achieve state-of-the-art camera pose estimation with comparable training time.

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

## A APPENDIX

### A.1 VISULIZED RESULTS OF SPARSE INPUT

We construct the mesh based on the estimated scene coordinates of the coarse stage and fine stage and visualize the camera pose estimation results in Figure 4. We may find that in the coarse stage, not only the pose estimation error is relatively large, but also the quality of the reconstructed details is low. In the refined stage, the performance is much better.

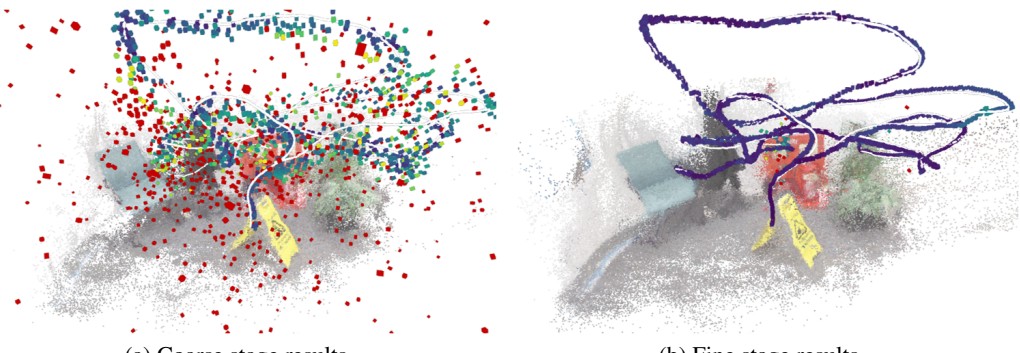

(a) Coarse stage results.                    (b) Fine stage results.

Figure 4: The localization results of the coarse-to-fine method for sparse view circumstances.

### A.2 VISULIZED RESULTS OF PoI

The visualized camera pose estimation results of 7Scenes are shown in Figure 5. The trajectories of the ground truth camera pose are drawn in white, while the color of the predicted trajectories is set according to the estimated translation error. As translation errors increase, the color tends to change from purple to red, following the color spectrum of the rainbow. To make the camera pose prediction results clearer, we also draw a mesh rendering view built from the estimated scene coordinates of the training data in the same frame for correspondence.

### A.3 ABLATION OF PoI

Table 4: Median errors of different implementations of PoI on 7Scenes and Cambridge dataset.

| Method | 7Scenes | | | Cambridge Landmarks | | |
|---|---|---|---|---|---|---|
| | trans↓ | rot↓ | $U_{5cm, 5\deg}$ ↑ | trans↓ | rot↓ | $U_{10cm, 5\deg}$ ↑ |
| base | 2.8cm | **0.8** | 36.5% | 17.7cm | **0.3** | 32.4% |
| base+poa | 4.6cm | 1.3 | 18.9% | 17.6cm | **0.3** | 32.2% |
| base+poi | **2.7cm** | **0.8** | **37.3%** | **16.6cm** | **0.3** | **33.1%** |

To evaluate the effectiveness of our PoI approach, we conducted some experiments on PoI in different settings. As shown in Table 4, We set the training process using only query data from the

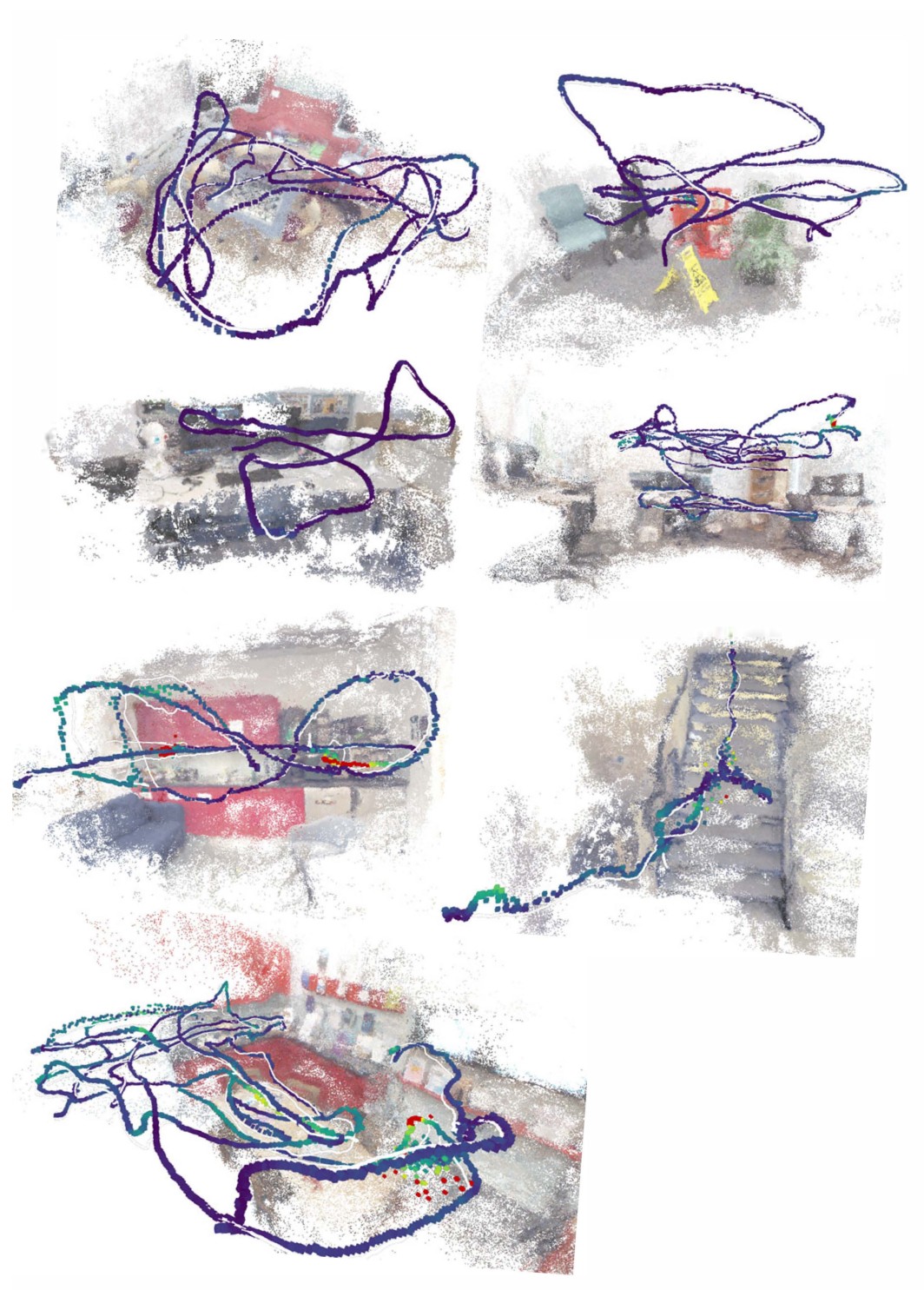

Figure 5: Visualized camera pose estimation results of 7scenes dataset.

training set as the **case 'base'**. In this case, the training setting is similar to Ace's. **Case 'base+poa'** indicates the training with data from the training set and all rendered pixels of the proposed novel pose rendering method. **Case 'base+poi'** is our final method, with the sampled novel poses and PoI algorithm. From the results, we may find that if we directly use sampled images with the training

data without filtering, the results will be far worse than the baseline. It is easy to understand that because the mapping process is filled with low-quality pixels, it would misguide the network.

