# OpenReview forum: "PoI: Pixel of Interest for Novel View Synthesis Assisted Scene Coordinate Regression"
_ICLR.cc/2025/Conference — ICLR 2025 Conference Withdrawn Submission_

### Official Review · Reviewer_EEE7 · 2024-10-23

**Soundness:** 2
**Presentation:** 2
**Contribution:** 2
**Rating:** 3
**Confidence:** 5

**Summary:**

This paper proposes to use novel view synthesis (NVS) techniques to augment the training of SCR methods. Considering the quality of rendered novel views, this paper also proposes a pixel-of-interest (PoI) filter to select well-rendered pixels and filter bad pixels.

**Strengths:**

1.	While this paper focuses on visual orientation, I believe it would be valuable to explore which areas of the rendered images from NVS techniques are well-rendered and which are not without ground truth reference. This could provide deeper insights into the 3D vision community.

2.	Although PoI offers only a marginal improvement over SCR methods when using all training data, it demonstrates significant potential for enhancing SCR accuracy with sparse input. This highlights the method’s value in visual localization tasks that rely on sparse reference images.

**Weaknesses:**

(1)	PoI does not keep the advantages of the latest SCR methods like ACE and GLACE which do not require a full 3D representation of the scene and simply require images and poses to train implicit neural networks in a very short time (around 20 minutes) to predict 2D-3D correspondences. The results suggest that PoI offers very marginal improvements to SCR methods but introduces significant overhead in mapping time. I don't see Table 2 clarifying the time spent in Figure 2 (a) for mapping.

(2)	The novelty of this paper is limited. Many papers have discussed the use of NVS techniques for data augmentation of visual localization methods. This paper uses off-the-shelf NeRF-W and 3DGS/MVSplat without exploring modifications to improve the robustness of rendered images or the generalization ability of sparse-view NeRF/3DGS. The fast training method of ACE is not a contribution to this paper.

(3)	There is a notable inconsistency in the mapping time for the Cambridge dataset. The original GLACE paper [5] reports a mapping time of 20 minutes, whereas Table 2 of this paper lists it as 3 hours.

(4)	Deadlock in Augmenting SCR Methods with NVS: The use of off-the-shelf NVS methods to augment SCR techniques creates a fundamental deadlock. Both SCR and NVS methods suffer from limited generalization to perspectives outside the training set. The core idea of the paper—enhancing SCR training using novel view rendered images—faces a challenge because NVS methods (e.g., NeRF, 3DGS) struggle to generate high-quality renderings for areas not fully covered during training. Consequently, SCR methods exhibit high errors on these frames in the test set. Meanwhile, for areas already covered by the training set, SCR has already learned sufficiently. This makes data augmentation using 3DGS-rendered images unlikely to significantly improve the accuracy of SCR methods.

(5)	The paper does not specify what kind of ground truth was used for training PoI on the 7Scenes dataset. Several papers have demonstrated the significant impact different GTs can have on results [1][2][3][4][5]. NeFeS highlights that the quality of NeRF-rendered images can be affected by the choice of GT. Based on Table 1, it appears this paper may have used inaccurate dSLAM GT for evaluation, which undermines the validity of the reported results.

(6)	This paper misses some important related work:
how to evaluate the rendered quality of novel view synthesis without ground truth references:

“CrossScore: Towards Multi-View Image Evaluation and Scoring” ECCV 2024.

How to avoid SCR approaches train on samples from all image regions:

“Reprojection Errors as Prompts for Efficient Scene Coordinate Regression” ECCV 2024

(7)	Minor: the paper is unpolished and a bit rough, with inconsistent terminology (e.g., "Nerf" and "Ace" should be consistently referred to as "NeRF" and "ACE"). Additionally, there are several incorrect bolded numbers in Table 1 (pumpkin, kitchen, Avg. colums).

[1] Brachmann, Eric, et al. "On the limits of pseudo ground truth in visual camera re-localisation." ICCV. 2021

[2] Chen, Shuai, et al. "Neural Refinement for Absolute Pose Regression with Feature Synthesis", CVPR 2024

[3] Trivigno, Gabriele, et al. "The Unreasonable Effectiveness of Pre-Trained Features for Camera Pose Refinement". CVPR 2024

[4] Brachmann, Eric, Tommaso Cavallari, and Victor Adrian Prisacariu. "Accelerated coordinate encoding: Learning to relocalize in minutes using rgb and poses." CVPR. 2023

[5] Wang, Fangjinhua, et al. "GLACE: Global Local Accelerated Coordinate Encoding." CVPR. 2024

**Questions:**

Given the marginal improvement introduced by PoI, I have a few suggestions:

1.	Could you provide the 5cm, 5° accuracy percentages (similar to Table 4) to demonstrate that adding PoI indeed improves the accuracy of ACE/GLACE when using all the training data?

2.    Could you reevaluate your method on 7Scenes using SfM GT over ACE/GLACE?

2.	Could you evaluate your method on additional datasets, such as 12Scenes, Wayspots and Aachen Day & Night, to showcase a more substantial improvement in accuracy over ACE/GLACE?

---

### Official Review · Reviewer_HzYN · 2024-11-03

**Soundness:** 1
**Presentation:** 2
**Contribution:** 1
**Rating:** 3
**Confidence:** 3

**Summary:**

This submission proposes a camera localisation method. The method is about utilising high-quality rendered pixels and their features from NeRF-W rendered views to assist recent Scene Coordinate Regression methods, such as ACE and GLACE.

Specifically, high-quality rendered pixels/features are thresholded via reprojection errors between rendered views and query views. While I enjoy the simplicity of the proposed method, the method section is slightly unclear to me, and the improvements compared with previous works are limited.

**Strengths:**

The research direction of utilizing NVS for camera localization is both meaningful and challenging. I appreciate the efforts being made to advance this field and I enjoy the simplicity of the proposed method.

**Weaknesses:**

1. Inaccurate description in L34-L38, "Traditional methods for camera relocalization can be categorized into two main types: Camera Pose Regression (CPR) methods and Scene Coordinate Regression (SCR) methods" -- there are many more types of camera (re)-localisation methods than these two types. CPR and SCR are recent network methods, and there are many classical epipolar geometry based methods using 2D-2D, 3D-3D, or 3D-2D information. On the network side, there are NetVLAD and Visual Place Recognition (VPR).

2. Missing method details in Sec 4.1. It is unclear how the pixel-of-interest or feature-of-interest are "combined". Questions listed below.

3. Marginal improvements over previous works. In Table 1 and 2, PoI performs similarly to previous works, leading to questions about the effectiveness of the proposed method.

**Questions:**

Sec4.1, L296, how does this "combine and shuffle" performed exactly? Specifically, both I_n and I_q denote a set of images, how does the re-projection performed? Since all images in I_n and I_q have different poses, the filtered features and the features for query images would not be aligned, how are they "combined and shuffled"?

---

### Official Review · Reviewer_ssiL · 2024-11-03

**Soundness:** 3
**Presentation:** 2
**Contribution:** 3
**Rating:** 6
**Confidence:** 3

**Summary:**

This paper presents Pixel of Interest--a method to improve camera pose estimation in scene coordinate regression  by integrating novel view synthesis data while addressing issues commonly associated with it, such as blurring and ghosting. These artifacts, whcich are particularly problematic for SCR methods, can compromise pose estimation accuracy due to the pixel-level precision required. To counter this, their method, PoI, introduces a filtering mechanism that adaptively selects only well-rendered pixels based on real-time reprojection loss during training, discarding low-quality pixels. Additionally, a coarse-to-fine variant of PoI enables scene coordinate regression from sparse data, ensuring adaptability in scenarios where training data is limited. Through extensive experiments on datasets including indoor (7Scenes) and outdoor (Cambridge Landmarks) settings, the authors demonstrate PoI’s ability to achieve SOTA performance across various tasks, with enhanced efficiency and scalability compared to other approaches.

**Strengths:**

The PoI method stands out by bringing novel view syntehsis to scene coordinate regression frameworks, which traditionally struggle with synthetic data due to the precision needed at the pixel level. While NVS-based data augmentation has proven effective in camera pose regression, its integration into SCR is novel and valuable, especially since SCR typically achieves higher localization accuracy. Their method's filtering mechanism, rooteid in pixel-level reprojection error, is an innovative solution for selectively improving training data quality. This originality is highlighted by the focus on pixel-level data augmentation and filtering, which addresses specific challenges of SCR, potentially broadening the applications of SCR. Furthermore, I believe their experimental rigor is impressive. The authors validate PoI’s effectiveness with quantitative and qualitative results on various benchmark datasets. The architecture is meticulously designed, particularly the reprojection error-based filtering mechanism, which minimizes the noise introduced by synthetic images, thus improving SCR models’ stability. They demonstrate PoI’s performance on both 7Scenes and Cambridge Landmarks, and thus the authors establish a strong case for its applicability to both indoor and outdoor environments. Additionally, the coarse-to-fine strategy is a noteworthy refinement that tackles sparse input scenarios, further enhancing PoI’s practicality and robustness. The paper’s structure is clear and logical, asnd every  section builds on the last to explain the method’s role, design, and application. Figures and tables, such as those comparing translation and rotation errors across different methods, provide a visual understanding of PoI’s performance, supporting the claim of SOTA results. The decision to use a thresholded reprojection error as the filtering criterion is well-motivated, as it aligns with SCR’s need for pixel-level accuracy. PoI’s introduction is a substantial step forward for SCR methodologies, especially regarding the integration of synthetic data for model enhancement. The paper’s focus on a plug-and-play like module that can be integrated with SCR pipelines is particularly valuable for applications requiring accurate localization but limited by real-world data constraints. In fields like autonomous driving and robotics, where the quality and quantity of training data directly impact pose estimation, PoI’s filtering approach offers a powerful way to leverage synthetic data without sacrificing accuracy. The authors show that the work also invites further research on pixel-level filtering in other tasks, potentially making PoI a foundational technique for data enhancement in SCR.

**Weaknesses:**

I believe the choice of reprojection error as the filtering criterion is effective I think it could also  benefit from a more in-depth discussion. The authors could explain why other potential metrics  were not considered, or if they were, why reprojection error was ultimately selected. Moreover, the paper does not address the threshold’s dynamic nature in great detail-- this could be crucial, as small changes in threshold values might impact model performance. A sensitivity analysis showing the effect of varying threshold values on model accuracy could clarify the robustness of PoI under different filtering settings. The paper mentions that incorporating a high proportion of rendered images in scene coordinate regression models can lead to model collapse; however, there is limited discussion on why this happens or how PoI’s filtering mechanism counters it. Since synthetic data could introduce noisy labels, a detailed analysis of these issues, backed by experiments with varying ratios of synthetic to real data, has the potential to strengthen the paper's argument. I believe it may also be beneficial to outline scenarios where PoI’s filtering may be less effective, such as in highly dynamic environments or under extreme lighting variations, to provide a more balanced view of its applicability. The authors present PoI’s use mainly in the context of non end-to-end scene coordinate regression models, which are less common than end-to-end methods in practical applications. While the paper mentions that PoI is challenging to integrate with end-to-end  models, it does not fully address *how* an adaptation might be achieved or the specific technical hurdles involved.

**Questions:**

1. How was the optimal reprojection error threshold chosen for filtering out poorly rendered pixels? Was it determined experimentally, or is there a theoretical basis for its selection? If you show results across a range of threshold values, it may  help in understanding the trade-offs between the number of filtered pixels and the overall accuracy
2. The pixel-level filtering process may introduce computational  chhallenges as dataset size and image resolution increase. Could you  comment on strategies  you have considered to improve computational efficiency, particularly for larger datasets?
3.  While your method leverages NeRF-W to handle changing lighting conditions and dynamic elements, it remains unclear how it performs in highly dynamic scenes with frequent changes. It may be worthwhile to expand on the limitations of PoI under such conditions, and whether incorporating more advanced novel view synthesis techniques or additional dynamic embeddings might improve performance in these challenging scenarios?
4. You have highlighted PoI’s success in non end-to-end scene coordinate regression, have you considered adapting it for end-to-end frameworks? If it were adapted for end-to-end SCR, what architectural changes would be required, and what might be the   challenges?

---

### Official Review · Reviewer_Zdau · 2024-11-08

**Soundness:** 3
**Presentation:** 3
**Contribution:** 2
**Rating:** 5
**Confidence:** 4

**Summary:**

The paper proposes a method of using artificially generated novel views to augment a scene coordinate regression-based visual (re)localiser.

**Strengths:**

The problem the paper targets is novel, as is the overall approach.
The paper is mostly well written and easy to understand.
The results demonstrate an improvement over standard ACE and GLACE.

**Weaknesses:**

The main weakness, in my view, is the lack of an ablation section. The paper makes certain decisions about threshold (e.g. the weight of the rendered features of 0.1 or the reduction of PoI weight) that, to the best of my knowledge, are not evaluated anywhere. Importantly, as I understand it, the actual importance of the artificial pixels / features is quite low, especially toward the end of the optimisation.

The improvement over ACE, while numerically there, is not very high (i.e often in the 1/0.1 range). This feels low, and potentially proportionate to the practically low importance of the artificial pixels / features. This should be discussed more in the paper. Importantly here, it would be interesting to know what is the variance of the results, and how this compares to e.g. ACE or GLACE -- this might dispel my worries that the numerical improvements are within noise levels.

The "mapping time" numbers are somewhat deceiving, as I do not believe they include the NeRF-W training and generation time, itself non-zero (and potentially, in practical, an order of magnitude above ACE/GLACE training).

**Questions:**

It would be interesting, and potentially very useful, if the authors were able to show that a random selection of artificial points (with their 0.1 weights and the scheduled reduction of PoI weight) produces worse results than their approach.

It would also be important for the authors to demonstrate that the the improvements are larger than the variability in ACE/GLACE scores.

---

### Note · Authors · 2024-11-13

I have read and agree with the venue's withdrawal policy on behalf of myself and my co-authors.